# Bystanders' attitudes towards drone delivered Automated External Defibrillators for out-of-hospital cardiac arrest: A qualitative interview study

Celia J. Bernstein[1], Christopher M. Smith[1]*, Carl Powell[2], Mary O'Sullivan[3], Mark Holt[3], Keith Couper[4], Nigel Rees[2]

1 Warwick Clinical Trials Unit, University of Warwick, Coventry, United Kingdom, 2 Welsh Ambulance Services University NHS Trust, Pre-Hospital Emergency Research Unit (PERU), Institute of Life Sciences Swansea University, Singleton Park, Swansea, Wales, United Kingdom, 3 Patient and Public Involvement (PPI) representative, 4 Critical Care Unit, University Hospitals Birmingham NHS Foundation Trust, Birmingham, United Kingdom

* c.smith.12@warwick.ac.uk

## Abstract

### Background

Early cardiopulmonary resuscitation (CPR) and defibrillation with Automated External Defibrillators (AEDs) by the public at an out-of-hospital cardiac arrest (OHCA) increases patient survival, but AEDs are infrequently used. Using drones to deliver AEDs may be one way to increase uptake, but there is limited understanding about what members of the public think about this. The aim of the study was to explore public attitudes towards drone AED delivery for OHCA.

### Methods

We conducted 14 remote, semi-structured interviews with real-life OHCA bystanders. Participants were recruited via social media, a UK cardiac arrest survivor charity and the Welsh Ambulance Services University NHS Trust. We analysed data using the Theoretical Domains Framework and mapped findings to the Capability, Opportunity and Motivation model of Behaviour (COM-B) to identify perceived barriers and facilitators to the retrieval and use of drone-delivered AEDs. We used The Behaviour Change Wheel to identify potential interventions to optimise use of drone-delivered AEDs.

### Results

Participants experienced varying levels of physical and social opportunities in relation to (un)available AEDs and (in)appropriate support from the call-handler, affecting the likelihood of them performing CPR and/or using an AED effectively. Most participants

**Data availability statement:** All relevant data are within the paper and its Supporting Information files.

**Funding:** This study was funded by the National Institute for Health and Care Research (NIHR) under its Research for Patient Benefit programme (Grant Reference Number NIHR204382) and Health and Care Research Wales (IRAS: 318417). The views expressed are those of the author(s) and not necessarily those of the NIHR, the Department of Health and Social Care or Health and Care Research Wales. The funders had no role in study design, data collection and analysis, decision to publish, or preparation of the manuscript.

**Competing interests:** Competing interests are declared in the manuscript: Christopher M Smith has volunteer roles with Resuscitation Council UK, European Resuscitation Council and the International Liaison Committee on Resuscitation. Keith Couper is chair of the National Institute for Health Research (NIHR) Research for Patient Benefit funding committee for the West Midlands region in the UK. Nigel Rees receives research and development funding from Health & Care Research Wales, who also provided UK NHS delivery funding for this project. Nigel Rees is Assistant Director for Research & Innovation for Welsh Ambulance Services NHS Trust.

**Abbreviations:** AED: Automated External Defibrillator; BCW: Behaviour Change Wheel; COM-B: Capability, Opportunity, Motivation Behavioural framework; COREQ: Consolidated Criteria for Reporting Qualitative Studies; CPR: Cardiopulmonary Resuscitation; OHCA: Out-of-Hospital Cardiac Arrest; TDF: Theoretical Domains Framework; UAV: Unmanned Aerial Vehicle; WAST: Welsh Ambulance Services University NHS Trust.

were unsure about how to use an AED, and none knew how a drone-delivered AED system would work in practice. Many participants questioned whether they would possess sufficient capability and motivation to retrieve and/or operate a drone-delivered AED during a time-critical emergency. There were five key themes for potential interventions: incorporating information about drone-delivered AED use into pre-existing training programmes and materials; ensuring drone use complies with specific regulatory and/or legislative requirements; making the drone-delivered AED easy to identify and access; optimising call-handler scripts to incorporate drone-delivered AED use; providing social support via a robust co-responder model to complement drone-delivered AED use.

## Conclusions

Participants accepted drone-delivered AEDs for OHCA, but were unsure if it would be effective. They identified several issues that we addressed through the development of a comprehensive intervention framework. A comprehensive call-handler script that incorporates drone-delivered AED use and support for bystanders was the most salient potential intervention for future testing by relevant stakeholders.

## Background

Globally, fewer than 10% of people survive to 30 days following out-of-hospital cardiac arrest (OHCA) [1,2]. Prompt defibrillation with publicly-available Automated External Defibrillators (AEDs) and high-quality cardiopulmonary resuscitation (CPR), which members of the public ('bystanders') can perform before the ambulance service arrives, is crucial for increasing OHCA survival with good quality of life [3,4]. However, AEDs are infrequently used. In England (2024), AEDs were used in 11.9% of bystander-witnessed OHCAs, whereas CPR was performed in 76.8% [5].

There are several barriers to successful bystander AED use, many of which relate to difficulties in locating or retrieving one. People often do not know where a nearby AED is [6] and may assume that it will be hard to reach [7]. AEDs are often not equitably or optimally placed [8]. Areas with high deprivation and a greater proportion of inhabitants from an ethnic minority background often have fewer AEDs compared to other areas [9,10]. Bystanders' concerns about causing a patient harm [11,12] mean that they will often remain at a patient's side, even if an AED is nearby [6,13].

One potential option to overcome some of these barriers is the use of Unmanned Aerial Vehicles (UAVs, 'drones') to deliver AEDs. Several Western countries have produced models of optimally-located drone bases to show how these could have delivered AEDs before an ambulance arrived in historical OHCA cases [14–18], and there are now operational drone-delivered AED networks [19,20]. In Sweden, a drone was deployed to 72 OHCAs over 13-months (April 2021 to May 2022), with AED delivery occurring before ambulance arrival in 37 cases and the AED attached

to the patient in six cases. An AED delivered a shock to two patients, with one surviving to 30 days [19]. In Denmark, a drone was dispatched to 16 eligible OHCAs (June 2022 to April 2023). An AED was delivered in all cases but none were attached to the patient, largely due to a swift ambulance service response [20].

Bystanders generally report positive experiences with interacting with a drone-delivered AEDs in simulation studies [17,21–23]. However, in our recent simulation study, bystander participants sometimes experienced difficulties with listening to and following instructions from the AED and emergency call-handler simultaneously, particularly when these instructions conflicted with each other [23]. Further, call-handlers in Sweden reported uncertainty about when to instruct bystanders to leave the patient to retrieve the drone-delivered AED and how long chest compressions should be paused to allow the bystander to retrieve it [24]. Call-handlers' communicative behaviours [25] and understandings of how to comply with guidelines related to AED retrieval and use [24] may affect the likelihood of an AED being retrieved and used during an OHCA.

There is limited in-depth study about what real-life OHCA bystanders think about the use of drone-delivered AEDs for OHCA. Thus, the aim of this study was to use a validated theoretical framework to explore the attitudes of real-life OHCA bystanders, who may or may not have used an AED, about drone-delivered AEDs and to use these findings to inform the development of novel interventions to optimise drone-delivered AED use.

## Methods

### Study design and setting

We conducted remote semi-structured interviews with members of the public who had previously acted in a 'bystander' or 'Good Samaritan' capacity during an OHCA.

We based the interview schedule (see Supporting information) on the COM-B framework. The interview questions focused on the key components of the cardiac arrest response, including recognising that the person was having a cardiac arrest, interacting with the emergency ('999') call-handler and performing bystander interventions (i.e., CPR with or without defibrillation). The interviewer (CJB) also asked the participant how they would feel about receiving an AED via drone during the cardiac arrest, including any potential challenges. CJB recorded the participant's age, gender and whether they had received any training in CPR and/or AED use either prior to and/or following the incident.

### Participant selection and recruitment

We initially recruited bystanders from Welsh Ambulance Services University NHS Trust's (WAST) and University of Warwick's institutional social media accounts, and via a WAST internal database of cardiac arrest bystanders (between 2nd April and 18th June 2024). Interested parties were asked to contact the research team via a study-specific institutional email for further information.

However, we needed to amend the recruitment strategy based on our suspicion that several individuals had volunteered to participate via our open social media recruitment approach, despite knowingly not meeting our eligibility criteria. Subsequently, WAST and Sudden Cardiac Arrest UK (a UK resuscitation charity, which supports over 600 individuals who have responded to or been affected by sudden cardiac arrest) internally disseminated information about the project to their diverse memberships. We recruited further between 3rd September and 1st October 2024.

Before the interview, we sent participants a Participant Information Sheet and Consent Form so they could review the documents and ask questions about the study if required.

We included people who provided informed consent to undertake an audio-recorded interview in English and self-reported that that they had provided CPR with or without the use of an AED at an OHCA. Participants could have a health-care background, but they must have responded to the OHCA being discussed in the interview in a bystander or Good Samaritan capacity. Participants received a gift voucher to thank them for their time.

## Data collection

Each participant was interviewed once using telephone or via Microsoft Teams. Only the interviewer and interviewee were present at the interview. Those who chose to have their interview via Microsoft Teams could opt to have their camera on or off. All interviews were audio-recorded and conducted in English.

CJB is a senior research fellow with a PhD and experience in qualitative health research and qualitative interviewing. She had no prior knowledge of any participant before consenting them to the study. She introduced herself as the senior researcher on the project and informed participants that her role was to listen to their experiences of responding to a real-life OHCA.

Audio recordings were transcribed verbatim by a commercial transcription service.

## The use of a theoretical framework

The Theoretical Domains Framework (TDF) has been applied in psychology and health services research to explore the barriers and facilitators to behavioural change in different health contexts [26,27]. The 14 domains from the TDF have been further distilled into the Capability, Opportunity and Motivation model of Behaviour (COM-B). This system is used to understand how the three underlying sources of behaviour – capability, motivation and opportunity – interact to predict and influence behaviour in a given situation [27]. These core behaviour targets can then be mapped onto the Behaviour Change Wheel (BCW) to identify potential interventions to enable the desired behaviour [28,29].

## Data analysis

We used a framework approach. Firstly, CJB summarised the main points from each interview and wrote detailed reflexive fieldnotes immediately afterwards. This helped to facilitate familiarity with the data on the perceived facilitators and barriers to drone-delivered AED use, and enabled CJB to reflect on her interviewing technique and consider the relevance of the dataset to the TDF and COM-B model.

Secondly, CJB read, anonymised and cross-checked each transcript against the audio-recording for accuracy, adding preliminary ideas and codes to the initial summaries. It was during this stage that it was noticed that some participants who turned their camera off during the interview had similar voices and/or provided almost-identical accounts. After discussion with the research team, we decided to exclude these responses from the analysis.

Thirdly, CJB imported the remaining interview transcripts into NVivo 15 for formal analysis by generating a modified coding framework based on the TDF. Studies have shown that TDF domains are broad theoretical categories that may prevent researchers from fully considering a range of behavioural determinants and influences [30]. Thus, to capture nuance in the data and illuminate the complexity of individual behaviour, we coded segments of interview text (e.g. sentences, paragraphs) to TDF domains and/or associated subcategories. These subcategories included a mix of inductive codes and deductive TDF constructs based on their relevance to the data. For example, interview data on bystanders' (un)willingness to do CPR and/or defibrillation were coded to the 'feeling morally obliged (not) to take action to avoid patient harm' inductive code within the broad 'Intentions' TDF domain, while data pertaining to disagreements between the bystander, onlookers and/or the ambulance service were coded to the 'inter-group conflict' TDF construct within the overarching 'Social Influences' TDF domain. Two inductive codes – 'admission to hospital' and 'experiencing distress during the interview' – were not mapped to any TDF domain but helped to contextualise the findings.

Fourthly, once all the interview transcripts were coded to the TDF domains and associated subcategories, we mapped the overarching TDF domains (together with these subcategories) to the components of the COM-B model (Table 1). Finally, in collaboration with CMS, who is a clinical academic with expertise in OHCA management, CJB generated a comprehensive narrative about bystanders' attitudes towards using a drone-delivered AED at the scene of an OHCA. Participants themselves were not asked to provide feedback on the findings.

Throughout the coding and analysis process, we adopted different validity procedures to bolster the credibility of the findings. We applied our different backgrounds and expertise to refine the coding framework [31], used an audit trail to

**Table 1. Mapping the TDF domains to the COM-B model.**

| COM-B component | COM-B sub-component | TDF domain |
|---|---|---|
| Capability | Psychological | Knowledge |
| | | Memory, attention and decision processes |
| | | Behavioural regulation |
| | Physical and psychological | Skills |
| Opportunity | Physical | Environmental context and resources |
| | Social | Social influences |
| Motivation | Reflective | Social/professional role and identity |
| | | Beliefs about capabilities |
| | | Optimism |
| | | Intentions |
| | | Goals |
| | | Beliefs about consequences |
| | Automatic | Reinforcement |
| | | Emotion |

Adapted from Michie et al 2014 [28].

document key messages and analytic decisions and reflected on how our different levels of familiarity with the topic may have shaped our interpretations and conclusions (researcher reflexivity) [32].

### Intervention development

We used the BCW to develop potential interventions around the effective implementation of drone-delivered AEDs. Firstly, we used COM-B to identify the target behaviour [29] getting bystanders to use a drone-delivered AED during an OHCA. Secondly, we mapped this behaviour to relevant intervention functions within the BCW [28,29]. Thirdly, we used the Behaviour Change Techniques Taxonomy version 1 [33] to identify suitable behaviour change techniques to modify the target behaviour. Fourthly, we identified how these interventions could be implemented in practice via suitable policy options and modes of delivery. Finally, we assessed the feasibility of the potential interventions in relation to practicability, affordability, safety and cost/effectiveness [28].

The richness of the data coupled with the subtly of the different intervention functions and synergistic relationship between the components of COM-B has necessitated the development of a comprehensive intervention framework (see Supporting information). Since these interventions would need to be tested and refined in concert with each other, we have grouped them into prominent interlacing themes (Table 3).

### Ethics approval and consent to participate

This study received NHS Research Ethics Committee (REC) approval from the Wales REC 5 committee (24/WA/0034) on 11th March 2024. We prospectively registered the study at clinical trials.gov (NCT06334718).

Participants provided either written informed consent prior to the interview, or verbal consent during the telephone or video-conference call, immediately before the interview. For those providing verbal consent, CJB recorded this by reading through the consent form and asking participants to answer 'yes' or 'no' to each question before the interview began. Participants were informed that anonymised transcripts from the interview could be made available for other researchers in the future, and as part of the consent process they explicitly agreed to the statement: "I understand that the research team may share anonymous information in my interview with other researchers to support their research studies and analyses in the future."

Interview consent forms or audio recordings of those providing verbal consent were kept separate from participants' answers to the main interview.

**Patient and public involvement**

Two Patient and Public Involvement representatives were involved in this study. They reviewed the study design and essential study documentation, including interview schedule, Participant Information Sheet and consent form, and have also contributed to the study's dissemination and research outputs.

**Reporting**

We report this study according to the Consolidated Criteria for Reporting Qualitative Studies (COREQ) [34].

## Results

We received 62 expressions of interest (April–October 2024). We categorised 24 as potential imposters: 13 who had used suspect email address names, blank subject lines and/or short introductions [35] and 11 *after* study participation (whose data we discarded). A further 24 did not reply.

We conducted 14 eligible interviews, at which point we determined we had obtained a sufficiently thorough and nuanced understanding of the topics presented [36]. This was determined by the interviewer (CJB) after discussion with CMS, and was based not only on no new codes being identified but also on agreement that we had a rich understanding of the topic at hand that was unlikely to be added to by additional recruitment. Interview duration ranged from 32 to 63 minutes. One participant experienced mild distress during the interview, but no participant terminated the interview early or withdrew their participation.

Seven participants were female, six participants were male, and one participant was transgender. Five bystanders (35.7%) reported using an AED during the incident, two of whom had a medical background, and one with no clinical training who used an AED brought to them by paramedics. Further participant information is in **Table 2**.

### Facilitating AED retrieval and use following delivery by a drone

We directly coded the interview transcripts to TDF domains or to subcategories of each of the TDF domains. Several data segments coded to multiple TDF domains and/or associated subcategories. Four TDF domains and associated subcategories mapped to physical and psychological capability (in COM-B), two to physical and social opportunity and eight to reflective and automatic motivation.

We identified 21 different behavioural determinants required for a bystander to retrieve and use a drone-delivered AED successfully. We also developed 25 potential interventions to achieve this: 11 targeting a bystander's physical and psychological ability to retrieve an AED from a drone, five targeting physical and social opportunity and nine targeting reflective and automatic motivation. We used the following intervention functions: enablement (21 times), education (nine), environmental restructuring (twice), persuasion (twice), incentivisation (once) and training (once) (see Supporting information).

Through our expertise and robust interpretation of the data, we found that many of the 25 potential interventions (see Supporting information) were similar or complementary to each other; this reflects the fact that data segments coded to multiple TDF domains and associated subcategories (and hence COM-B and BCW components). Previous research has suggested that modifying the TDF in this way may generate a comprehensive and nuanced view of behaviour [30]. Therefore, we have condensed these interventions into five key themes reported below and discussed the findings from the interview study and BCW analysis in tandem within each theme. **Table 3** shows how each theme links to the relevant part(s) of COM-B, and the relevant intervention function(s) and policy category(ies) of the BCW.

### Theme 1: Incorporating information about drone-delivered AED use into pre-existing training programmes and materials for out-of-hospital cardiac arrest management

This theme principally relates to a bystander's (perceived) psychological capability to retrieve and use a (drone-delivered) AED. It also links to their (anticipated) state of mind (automatic motivation) and (un)willingness to do CPR and/or defibrillation (reflective motivation) during the real or imagined time-pressurised emergency.

**Table 2. Participant information.**

| ID | Gender | Age (years) | Prior involvement in another out-of-hospital cardiac arrest? | Training in CPR/AED? | Where did the cardiac arrest(s) discussed in the interview happen? |
|---|---|---|---|---|---|
| P1 | Male | 29 | No | CPR only (< 1 year pre-incident) | Public space – indoor |
| P2 | Transgender | 25–27 (self-reported) | No | No | Public space – indoor |
| P3 | Female | 28 | No | CPR only (one year post-incident) | Patient's house |
| P4 | Female | 25 | No | CPR only (10 years pre-incident). | Public space – outdoor |
| P5 | Female | 56 | No | CPR/AED (several, yearly) | Patient's and bystander's house |
| P6 | Male | 61 | Yes (retired healthcare professional) | CPR/AED (~10 years pre-incident) | Public space – outdoor |
| P7 | Female | 42 | Yes (Community First Responder) | CPR/AED (several, including <1 year pre-incident) | Public place – outdoor |
| P8 | Male | 28 | No | No | Public space – outdoor |
| P9 | Male | 26 | No | No | Public space – indoor |
| P10 | Female | 33 | No | CPR only (>15 years pre-incident) | Patient's and bystander's house |
| P11 | Female | 75 | No | CPR/AED (several, including <1 year pre-incident) | Patient's and bystander's house |
| P12 | Male | 27 | No | CPR only (pre-incident) CPR/AED (post-incident) | Public space – indoor |
| P13 | Male | 52 | Yes (healthcare professional and Community First Responder) | CPR/AED (several, including <1 year pre-incident) | Public space – indoor |
| P14 | Female | 61 | Yes (healthcare professional) | CPR only (several, including <1 year pre-incident) CPR/AED (post-incident) | Patient's and bystander's house |

Most bystanders without prior experience in cardiac arrest management had limited understanding about how to deliver CPR and/or use an AED:

'[…] you hear of defibrillators, but you think, "Oh, I don't even, you know, I'm not aware what they are," or... To be honest, you, you vaguely know, but you're not, you don't know enough about them […]' [P11]

Lacking capability also caused several participants to become emotionally distressed during their cardiac arrest experience and deliberate on the most suitable course of action:

'I really tried my best that day. I, I did. I was scared. I was, I was so scared. I, it was this fear. I had this fear, 'cause I've never experienced such a thing […] and seeing a lifeless body on the floor without knowing what to do … and then it just occurred to me that, "This, this could be a cardiac arrest. Why not give him [patient] a CPR to see if he could respond to it?"' [P3]

Additionally, two of the participants with healthcare training and who had prior real-life experience of cardiac arrest management were diffident about assisting unexpectedly in a new environment:

'It's, it's all very well doing your training and practising and being shown how to use them [defibrillators], but when, when the actual reality hits, it's still slightly, makes you slightly anxious […]' [P6]

Table 3. Potential interventions to enable drone-delivered AED retrieval and use.

| Inductive theme/ potential intervention | Sub-theme/ component of the potential intervention | Capability, Opportunity and Motivation (COM-B), Intervention function and Policy category |
|---|---|---|
| 1. Incorporating information about drone-delivered AED use into pre-existing training programmes and materials for out-of-hospital cardiac arrest management. | Provide resources for the public about where drone-delivered AED use fits within the chain of survival and how it can be successfully implemented. | **COM-B:**<br>Capability (psychological)<br>**Intervention function:**<br>Training<br>**Policy category:**<br>Guidelines |
| 2. Ensuring drone use complies with UK-specific regulatory and/ or legislative requirements. | Comply with UK-specific regulatory and/or statutory requirements regarding:<br>• Establishing operational and clinical parameters for initiating drone flight;<br>• a drone's payload;<br>• the operation of a drone;<br>• drone delivery planning, and;<br>• the use of a potentially life-saving medical device. | **COM-B:**<br>Opportunity (physical)<br>**Intervention function:**<br>Enablement<br>Environmental restructuring<br>**Policy category:**<br>Guidelines<br>Environmental/social planning<br>Regulation<br>Legislation |
| 3.Making the AED easy to identify and access | Ensure that the box with the AED inside is identifiable and has a safe easy-open mechanism. | **COM-B:**<br>Opportunity (physical)<br>**Intervention function:**<br>Environmental restructuring<br>**Policy category:**<br>Regulation |
| 4. Optimising the call-handler script to incorporate drone-delivered AED use into the management of an out-of-hospital cardiac arrest Considered the most salient theme. | Prepare the bystander for what is going to happen when they need to reach the AED during the incident. In particular, the call-handler will need to tell the bystander:<br>• That a drone will be dispatched;<br>• what the drone will look like;<br>• that the drone will deliver an AED to the ground;<br>• what the AED will look like; and,<br>• that they will be given specific instructions about how to retrieve the drone-delivered AED. | **COM-B:**<br>Capability (psychological)<br>Motivation (reflective)<br>Opportunity (physical)<br>**Intervention function:**<br>Enablement<br>Education<br>**Policy category:**<br>Guidelines |
| | Provide specific instructions (via phone and/or video) about how to:<br>• Use a drone-delivered AED safely and effectively including supporting the bystander to locate, identify, access and operate the device;<br>• access brief, pre-recorded video demonstration(s) about CPR and/or drone-delivered AED use;<br>• deliver CPR and/ or use a drone-delivered AED locating, identifying, accessing and operating the device by sending brief, pre-recorded video demonstration(s); and,<br>• manage other aspects of the cardiac arrest response so that the bystander has the physical and psychological capability to retrieve and use the drone-delivered AED. | **COM-B:**<br>Capability (physical and psychological)<br>Opportunity (physical and social)<br>Motivation (reflective)<br>**Intervention function:**<br>Enablement<br>Education<br>Environmental restructuring<br>**Policy category:**<br>Guidelines<br>Service provision |
| | Justify the importance of defibrillation and explain that the bystander is allowed to use the drone-delivered AED. Explain the importance of the bystander leaving the patient to get the drone-delivered AED if they are reluctant to do this. | **COM-B:**<br>Motivation (reflective)<br>**Intervention function:**<br>Enablement<br>Education<br>Persuasion<br>**Policy category:**<br>Guidelines |

*(Continued)*

**Table 3.** (Continued)

| Inductive theme/ potential intervention | Sub-theme/ component of the potential intervention | Capability, Opportunity and Motivation (COM-B), Intervention function and Policy category |
|---|---|---|
| | Provide emotional support to the panic-stricken bystander. Offer praise regularly and assure them that their safety has been considered. | **COM-B:** Motivation (reflective and automatic) **Intervention function:** Enablement Incentivisation **Policy category:** Guidelines |
| 5.Providing social support via a robust co-responder model to complement drone-delivered AED use | Support the bystander to safely, competently and efficiently retrieve and operate the drone-delivered AED. | **COM-B:** Capability (psychological) Opportunity (social) **Intervention function:** Enablement **Policy category:** Service provision |

Despite experiencing disruptive negative reactions, all participants took either principal (majority) or shared (minority) responsibility for the incident because they felt that they had a moral obligation to save the patient's life.

Only two participants (both Community First Responders (CFRs)) reported high self-efficacy in AED use. Yet, none of the participants were sure how drone-delivered AEDs would work. Some wondered whether a lack of knowledge about this technology would make it difficult for the bystander to know what to do:

CJB: […] 'how do you imagine drone-delivered defibrillation works?'

P2: 'I don't know. Like, actually it's something that would be good. Because from the little I know about drones, I feel like they are faster […] someone who don't really have knowledge on how to use it […] might have difficulties.'

The findings reveal a need to increase bystander awareness of and proficiency in the drone-delivered AED process before a cardiac arrest, by including accurate and accessible information about how it would be successfully implemented as part of response to cardiac arrest.

### Theme 2: Ensuring drone use complies with UK-specific regulatory and/or legislative requirements

This theme principally relates to physical opportunity, although the lack of awareness around where AEDs were located and perception among most participants that these were inaccessible at the time of the incident also corresponds to psychological capability and reflective motivation, respectively.

Some participants did not know the location of a nearby AED while managing the incident and had still been unsure at the time of interview:

'I wouldn't even know where our nearest one [defibrillator] is even now.' [P11]

If the defibrillator was deemed inaccessible, the call-handler advised the bystander to focus their efforts on delivering CPR:

'They [call-handler] said, "Oh, do you know if there's a defibrillator nearby?" "Oh, it's about half a mile away." And they said, "Okay, don't worry. Just carry on with the CPR."' [P10]

Several participants felt that they would have used an AED if one had been available. As such, some suggested that using a drone to deliver an AED to the scene could facilitate AED use:

'[…] it [a drone-delivered AED] comes to you, rather than you 'aving to go to it in a crisis situation.' [P13]

However, some participants felt that external issues related to the weather, poor drone delivery planning and/or the functioning of the drone battery could affect the drone's ability to deliver the defibrillator to the scene of a cardiac arrest.

Thus we have identified the need for drone-delivered AED systems to comply with different UK-specific regulatory and/or statutory requirements and to establish appropriate parameters to initiate drone flight. This includes both clinical parameters such as the location of a public-place AED, presence of multiple bystanders, time since patient collapse, ambulance response time; and safety parameters such as weather, visibility, restricted airspace on route to cardiac arrest location.

### Theme 3: Making the drone-delivered AED easy to identify and access

This theme principally corresponds to physical opportunity. It also aligns with reflective motivation given that participants imagined what might happen when the bystander tried to retrieve the AED.

Two participants (both healthcare professionals) felt that panic-stricken bystanders unfamiliar with how to manage a cardiac arrest might struggle to open a carry-case containing the AED. They were also concerned that this might increase bystander stress and cause delays to defibrillation:

'And if you drop it [drone-delivered AED] in the middle of a field, it's gotta be a bright orange box, about this big, flashing at them [bystander]. And then the question's gonna be, "Well, how do I open the box? How do I get in? What do I do with the box? What do I do, what do I do with the defib?" […] the, the drones dropped, dropped the defib by the front door […] and don't identify it, because it's, it's, it's, presumably, in a cosy box. You then get the box, and bring it back, and don't realise you've gotta open the box to, kind of, get the defib, to, kind of, open the defib.' [P7]

Therefore, we recommend that the AED carry-case is identifiable and can be opened safely and easily.

### Theme 4: Optimising the call-handler script to incorporate drone-delivered AED use into the management of an out-of-hospital cardiac arrest

This theme encapsulates all components of the COM-B model and the multiple challenges facing bystanders who need to effectively retrieve and use a drone-delivered AED.

Most participants felt unprepared when managing their cardiac arrest incident. Some found chest compressions were unexpectedly tiring, which could potentially affect their physical ability to retrieve a drone-delivered AED. Others who used an AED during their cardiac arrest incident, with no prior cardiac arrest experience, were overwhelmed and mentally fatigued from having to execute multiple tasks at once, including listening to AED voice instructions whilst interacting with the call-handler:

'And I, I think at the point, without their help, the people that were there, I felt, I felt a bit stretched thin. It's like trying to talk to someone, and try to do the lifesaving act at the same time. It just felt a bit clumsy […] I think it's just the reality of the situation. It's like, I felt like I had to be in two places, listen to the instructions, and get them right. Then apply them, so my uncle doesn't die.' [P4]

Despite recognising the benefits of using an AED, some participants were concerned that the anxiety they had felt during their incident might impact on their ability to use one effectively in the future:

'Because, you know, although on one hand I said I was running on adrenaline, you know, having to do anything [such as defibrillation] that might have been slightly fiddly, I, you know, I probably shaking.' [P10]

Most participants recognised that drone-delivered AED could expedite the delivery of a defibrillator to the scene of a cardiac arrest. One participant without prior experience also thought that bystanders might feel reassured if they saw a flying drone arriving with the potentially life-saving defibrillator:

'So that, that hope alone that the drone is coming or the defibrillator is close to you will give you this, kind of, courage to just keep doing the compression, chest, the, and the rest of those.' [P8]

However, though participants sometimes reported positive attitudes towards (drone-delivered) AEDs, most said that they would be reluctant to leave the patient's side to reach the device if they were acting alone in case the patient deteriorated or died while they were gone:

'Well, to be honest, that, that does make me a bit uncomfortable [leaving the patient if they are by themselves to get the defibrillator from the drone]. I, I would prefer if there's someone else that can get the defibrillator, instead of me having to leave my uncle. I, I don't think I would have been able to do that.' [P4]

'Cause what if the moment I leave, this person dies or something?' [P8]

Some participants reported not retrieving a nearby AED during their cardiac arrest situation because of concern that they lacked sufficient knowledge to use it effectively, thereby wishing to delegate this task to trained healthcare professionals instead:

'I didn't actually use the defibrillator […] it would be a lot more good if, if there were maybe some, someone in the medical field that was present, that maybe I'm just assisting […] my, my, my thought was, "Well, what if something go wrong?" […] actually, one that I'd just been to that that there's, there's one nearby, but I, I didn't, didn't let them bring it over to me […]' [P1]

Additionally, one of the CFR participants referred to ambulance service policies to justify a decision to remain with the patient:

'I wouldn't be leaving a patient. So, whether, whether as bystander, or not […] as volunteers CFRs, we, we are taught, you know, we, we don't go and get a defib if, if we're by ourselves.' [P7]

However, some bystanders felt that it would be reasonable to leave if the patient remained in sight:

'I would like it [drone-delivered AED] to be delivered very close to me (chuckles) again, if you're on your own […] you would have to leave your patient […] you could probably still see the patient when you were going to get the defibrillator.' [P11]

Others said that they would leave if they felt that they could reach the device in a relatively short amount of time:

'Well, it's, it's, it's the balance of odds really. If you don't leave them [patient] and get the defibrillator, they're probably not going to survive because all you're doing is, in terms of CPR, is, is keeping is keeping things going until you can attempt

defibrillation. So, as long as you didn't have to leave for particularly long I guess, I guess you'd, you'd have to leave if you were on your own. Yeah, you're balancing, balancing odds against time really, aren't you, balancing, balancing risk.' [P6]

Yet, some remained concerned that delivering to a supposedly nearby location such as the garden or street door might still be too far for bystanders to reach, particularly if they had to go down multiple flights of stairs or run to the end of a garden and back.

Another crucial factor potentially affecting the target behaviour was participants' attitudes towards the call handler. Most bystanders, with or without medical training, depended considerably on the expertise of the emergency call-handler to provide instructions on how to manage the cardiac arrest and became distressed when this support had been insufficient or lacking:

'But I, I do remember not being given any instructions [from the call-handler], no, "Put him on the floor." No, "Let me count you through it. Now do this, now do..." Nothing. Nothing, nothing, nothing.' [P14]

While there were mixed responses around how helpful the call-handler had been during the actual incident ('I think they [call-handler] did a remarkable job! […] they told me exactly what to do.' [P10]), most felt that the call-handler would be instrumental in drone-delivered AED use. Participants said that the call-handler should say when the drone is dispatched, when it is likely to arrive, its precise delivery location and how the bystander can identify and interact safely with the drone and the AED:

'If they [call-handler] could say, you know, "We can get a defib off, delivered to you by drone," and, kind of, the time response on that, then, yeah, I'd, I'd 100% be giving them that information and answering their questions to, to get that […] so if the drone was coming […] then, yeah, I'd want to know when it's coming.' [P12]

Most participants believed that the call-handler could bolster bystander motivation around reaching and using a drone-delivered AED by providing such practical support.

However, one participant said that she would pointedly refuse to retrieve the drone-delivered AED if she deemed the incident location unsafe:

CJB: 'What about if the call handler said, "You can get it over the road." Would you? And you're on your own.

P14: No. […] if I'm on my own, no.

CJB: You would override them in that scenario, yeah.

P14: Yeah, because what you do about your front door shutting behind you? Can't get back in your house, it's the middle of the night, you're in a nightdress. You're in a really crappy area. How does that all work? No.'

These findings suggest that the accessibility of the drone-delivered AED and a moral imperative to help the patient might not be sufficient to ensure its successful retrieval and use. It would be critical for the call-handler to facilitate support by providing advice when required during the call. The call-handler would need (if deemed appropriate) to encourage a lone bystander to leave the patient's side. There are several components of an emergency call script where the call-handler's words could impact on the successful retrieval and use of an AED (see **Table 3**).

## Theme 5: Providing social support via a robust co-responder model to complement drone-delivered AED use

This theme principally maps to a bystander's psychological capability and whether they have the social opportunity to retrieve a drone-delivered AED during a cardiac arrest.

Sharing responsibility for the management of the cardiac arrest increased bystander motivation and mitigated the risk of overexertion. One participant found it beneficial to retrieve a nearby AED while the other bystander continued to deliver CPR:

'We then realised he had stopped breathing, so we rolled him onto his back and the gentleman I was with started with chest compressions, whilst I went back out to the reception area to grab the on-site defibrillator […] it made me feel supported [to have another bystander there] […] so, yeah, I think we both kind of were able to support each other.' [P12]

However, most participants had managed the incident on their own. A minority had physically stopped disruptive people from impeding their efforts to defibrillate the patient at the scene or had spent valuable time justifying their need for an AED:

'I continued to ask for the defibrillator, and they [security guards] demanded to know who I was. I said, "I'm a qualified nurse. Obviously, I've responded to this incident. This is my findings." They said, "Well, she looks alright now" […] when, not only were the security denying me the defib, wanting to see ID, they also were disputing that she'd been in cardiac arrest.' [P13]

Consequently, another participant with prior experience wondered whether introducing a novel technology into a (crowded) public space would increase distraction. Other participants felt that an absence of human expertise by delivering an AED by drone might be detrimental:

'I feel like if it's delivered by a human, the person can literally explain and help facilitate the process of helping that person in an emergency situation. But if it's a drone, a drone cannot talk, so there, there would be a delay in this person coming to learn how to use it.' [P2]

As such, one participant said that he would find it easier to use a drone-delivered AED 'if maybe an assistant also would be present' [P1].

Further, there was strong consensus that participants would be willing to leave the patient to get the AED if they were fully supported by another bystander throughout the management of the OHCA:

'If there was another person, then I would go get it [drone-delivered AED]. I would go out to get it myself. 'Cause I know there's someone there with him [patient].' [P3]

The data thus suggests the benefit of having at least on other competent and co-operative person at the scene to increase the chances of a drone-delivered AED being retrieved successfully.

## Discussion

### Main findings

Getting an AED from a drone to a bystander and then attached to a patient is a complex process. A fully operational drone-delivered AED system needs to carefully consider the experience and needs of OHCA bystanders. Increasing knowledge about the use of drone-delivered AEDs in training materials may help prepare people. A drone-delivered AED also needs to be visible to bystanders. Since bystanders may doubt their capability or lack the motivation to retrieve and use a drone-delivered AED, the role of the emergency call-handler in preparing the bystander and motivating them to leave the patient to do this is key. Lone bystanders may need particular support and so a parallel focus on getting co-responders to the scene may also be beneficial.

### Relationship with previous literature

Drones may be one way to overcome the issue of infrequent bystander AED use during OHCA that is acceptable to bystanders [23,37]. Real-world studies in Sweden [19] and Denmark [20] have shown it can be done safely and

effectively. In a recent simulation study, we reported that bystanders felt a sense of relief when the drone delivered a potentially life-saving device close to the incident location [23]. Other researchers have also found that using drones for civil and health applications often correlate with higher levels of public support for this technology than if used for private and commercial uses [38]. Similarly, most participants in this interview study positively appraised the notion of AED delivery by drone for cardiac arrest and felt that it could be done quickly.

However, participants anticipated several problems with drone-delivered AED use. Firstly, most lacked familiarity with drones and AEDs, and all were unsure about how the process would work. During the actual incident, some participants had been concerned that their lack of confidence to locate and/or use an AED would cause patient harm. However, it is worth noting that most participants lacked confidence in CPR but still did it during the incident, with or without input from the call-handler. Other researchers [6,12] have reported that being legally responsible for potentially harming the patient and feeling guilty about causing patient harm, in conjunction with a paucity of knowledge in and low self-efficacy with locating and using an AED, were most likely to affect AED use among the public in England [6] and the Netherlands [12] – more so than the availability of AEDs themselves.

Secondly, participants expressed reluctance to leave a patient to retrieve the drone-delivered AED, particularly if managing the incident on their own. However, all participants reported that they would willingly retrieve the AED if they had face-to-face practical support from another bystander. In our recent simulation study, we reported uncertainty among bystanders about when to leave the patient [23]. In our current study, participants said that they would rely on the expertise of the call-handler to make this decision. However, in the operational system in Sweden, emergency call-handlers were also uncertain about the most appropriate time for the bystander to leave the patient to retrieve a drone-delivered AED, and how long they could spend away from the patient [24].

Merely increasing the availability of AEDs is not enough to improve utilisation; we must also address the multiple reasons underlying people's fears and motivations for (not) retrieving and using an AED [6]. Further, there is no guarantee that an AED will be used if available, which has been demonstrated in studies reporting the use of AEDs in both public [39] and home [40] settings.

### Implications for policy and practice

While the prompt use of an AED by a bystander can increase survival beyond hospital discharge to up to 70% if used within the first 3–5 minutes following collapse [4], rates of bystander defibrillation remain low [5,6,12,41]. However, prompt and high-quality chest compressions during CPR with minimal interruptions are also important to survival [3]. To address bystanders' concerns about the competing priorities of performing CPR and leaving the patient to retrieve an AED there must be clear rules about when to dispatch a drone and when a bystander can leave a patient. The call-handler using a standardised multi-composite script incorporating drone-delivered AED use would facilitate rapid retrieval of an AED – of particular importance when there is a lone bystander. Previous work has demonstrated that how the call-handler frames questions about (public-place) AED availability can affect the likelihood of a bystander retrieving it [25]. Consequently, the script would need to comprehensively address the following core components: 1) prepare the bystander for drone-delivered AED use ('the what'); 2) provide specific instructions about how to retrieve the device ('the how'); 3) justify the importance of defibrillation and the need to leave the patient to get the defibrillator ('the why'); and, 4) provide emotionally supportive and reassuring statements that acknowledge the wellbeing of the bystander throughout the interaction (see **Table 3**).

### Strengths and weaknesses

This is the first study to provide a rigorous behavioural analysis of the factors affecting the retrieval and use of a drone-delivered AED by bystanders during an OHCA. We have developed different interventions for stakeholders to test to optimise drone delivery of AEDs.

The sample size in this study is small and the participants might not be representative of the general population. It may have been that more motivated or confident bystanders were more likely to put themselves forward as participants. Certainly, a higher proportion of our participants reported CPR/AED training than the general UK population [42].

Further, these findings may not be fully applicable outside the UK. Whilst bystanders in simulation studies internationally generally respond positively to drones delivering AEDs [17,21–23], the delivery model in the UK may vary from those in other countries. Reasons for this may include differences in aviation regulations, existing AED availability and ambulance response times. This potentially limits the applicability of our findings to other countries.

We advertised to the study via institutional social media accounts of the researchers' institutions, as well as via their internal channels. However, we revised our approach after several possibly fraudulent responses. We believed that social media was the most likely source of these responses [43], so we modified our approach to advertise only via 'closed' channels. Nonetheless, determining whether or not participants were genuine was subjective [35], and so some uncertainty must remain about participants whose data *has* been analysed. If we have erred in our decisions to exclude participants based on suspected fraud, this may impact the validity of our results.

Social media recruitment can also limit the sample to those who are younger and more digitally literate [44]. Indeed, most participants (n = 8) were in their 20s or 30s at the time of interview and the majority (n = 9) had also assisted with a cardiac arrest in public, despite most OHCAs happening at home [5], affecting the generalisability of our findings.

### Implications for further research

Researchers need to assess the useability, acceptability and feasibility of the potential interventions by relevant stakeholder groups (e.g. ambulance services, call-handlers, co-responders) to ensure optimal provision of a drone-delivered AED network. It is important that any drone-delivery system that is implemented at scale is fully integrated into the community and ambulance service response to out-of-hospital cardiac arrest, which will ensure the best chance of improved patient outcomes [45].

It became apparent during the interviews that willingness to leave the patient's side would be a vital consideration in drone-delivered AED use. If bystanders are reluctant to retrieve the AED due to fears that the patient may come to harm while they are gone then getting the AED quickly may have limited benefit. Therefore, key interventions developed here focus on the role of the call-handler in knowing when to instruct or encourage the bystander to leave the patient's side if they are reluctant to do so. Since existing research indicates that call-takers do not always know how to comply with telephone-assisted cardiac arrest routines for drone-delivered AED use [24], our findings suggest the need to use the experiences of bystanders and call-handlers when developing a multi-composite standardised script for drone-delivered AED use to ensure compliance to it.

### Conclusions

All participants experienced negative affectivity to various degrees while managing an OHCA. Most participants also felt that they lacked capability to execute tasks and/or hadn't been (sufficiently) supported by the call-handler. The idea of drone-delivery of AEDs was positively received by most bystanders and considered potentially useful. However, there was uncertainty around how this process might be effectively implemented. It was also reported that introducing two relatively unfamiliar objects – drones and AEDs – into an already stressful situation could heighten the intensity of the bystander's negative emotional responses to the incident, particularly if they were acting alone and felt conflicted about whether to pause CPR to get the AED or remain with the patient.

Consequently, we have developed a set of potential interventions to address these issues. The development of a standardised and comprehensive call-handler script is required to facilitate the optimal implementation of a drone-delivered AED system for OHCA.

## Supporting information

**S1 Table. Possible barriers and facilitators to lay bystander retrieval and use of a drone-delivered defibrillator.**
(DOCX)

**S2 Table. Potential interventions for drone-delivered defibrillator retrieval by an out-of-hospital cardiac arrest bystander.**
(DOCX)

**S1 File. Topic guide.**
(DOCX)

**S2 File. Iterative interview questions (developed during the process of data collection).**
(DOCX)

**S3 File. Anonymised interview transcripts.**
(DOCX)

**S1 File. COREQ Checklist.**
(PDF)

## Acknowledgments

Carla Jones and Lauren Clarke at WAST for identifying and approaching OHCA bystanders.

## Author contributions

**Conceptualization:** Christopher Matthew Smith.

**Data curation:** Celia J Bernstein.

**Formal analysis:** Celia J Bernstein.

**Methodology:** Celia J Bernstein, Christopher Matthew Smith, Carl Powell, Mary O'Sullivan, Mark Holt, Keith Couper, Nigel Rees.

**Project administration:** Christopher Matthew Smith.

**Resources:** Celia J Bernstein, Carl Powell.

**Supervision:** Christopher Matthew Smith, Nigel Rees.

**Writing – original draft:** Celia J Bernstein.

**Writing – review & editing:** Christopher Matthew Smith, Carl Powell, Mary O'Sullivan, Mark Holt, Keith Couper, Nigel Rees.

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
