## [Decision Letter · Decision Letter 0]

23 Sep 2025

Dear Dr. Smith,

Thank you for submitting your manuscript to PLOS ONE. After careful consideration, we feel that it has merit but does not fully meet PLOS ONE’s publication criteria as it currently stands. Therefore, we invite you to submit a revised version of the manuscript that addresses the points raised during the review process. 

We look forward to receiving your revised manuscript.

Kind regards,

Nik Hisamuddin Nik Ab. Rahman

Academic Editor

PLOS ONE

Journal Requirements:

3. In the online submission form, you indicated that the anonymised transcripts generated during the current study and our analyses of them are stored securely at the host institution and are available from the corresponding author on reasonable request. Due to the risk of participant identification we will not share interview recordings.

4. Thank you for stating the following in the Competing Interests/Financial Disclosure * (delete as necessary) section:

Christopher M Smith has volunteer roles with Resuscitation Council UK, European Resuscitation Council and the International Liaison Committee on Resuscitation.

Keith Couper is chair of the National Institute for Health Research (NIHR) Research for Patient Benefit funding committee for the West Midlands region in the UK.

Nigel Rees receives R&D funding from Health & Care Research Wales who also provided UK NHS delivery funding for this project. He is also Assistant Director for Research & Innovation for Welsh Ambulance Services University NHS Trust.

We note that you received funding from a commercial source: [Name of Company]

This study was funded by the National Institute for Health and Care Research (NIHR) under its Research for Patient Benefit programme (Grant Reference Number NIHR204382) and Health and Care Research Wales (IRAS: 318417). The views expressed are those of the author(s) and not necessarily those of the NIHR, the Department of Health and Social Care or Health and Care Research Wales.

6. Please amend your authorship list in your manuscript file to include author Christopher Matthew Smith, Celia J Bernstein, Carl Powell, Mary O'Sullivan, Mark Holt , Keith Couper, Nigel Rees.

7. Please amend the manuscript submission data (via Edit Submission) to include author Bernstein, C J, Smith, C M, Powell, C, O’Sullivan, M, Holt, M, Couper, K, Rees, N.

8. Please remove all personal information, ensure that the data shared are in accordance with participant consent, and re-upload a fully anonymized data set.

Reviewers' comments:

Reviewer's Responses to Questions

**Comments to the Author**

1. Is the manuscript technically sound, and do the data support the conclusions?

Reviewer #1: Yes

Reviewer #2: Yes

2. Has the statistical analysis been performed appropriately and rigorously?

Reviewer #1: N/A

Reviewer #2: Yes

3. Have the authors made all data underlying the findings in their manuscript fully available?

Reviewer #1: No

Reviewer #2: No

4. Is the manuscript presented in an intelligible fashion and written in standard English?

Reviewer #1: Yes

Reviewer #2: Yes

Reviewer #1: The investigators explored public attitudes towards drone-delivered AEDs for OHCA by conducting 14 semi-structured interviews with real-life OHCA bystanders. They identified 5 themes and devised interventions.

1. The sample size is small and it is unclear whether the subjects are representative of the general population. Nevertheless, the study provides clues on the complexity of the process. I suggest that the authors acknowledge the small sample size in the limitations section.

2. The availability of AED doesn’t guarantee it being used. I would cite the Home use of AED trial (HAT), which did not improve the OHCA outcomes (DOI: 10.1056/NEJMoa0801651). One of the reasons for failure was insufficient use of the AEDs, despite resuscitation training, including AED, provided to the caregivers in the study.

3. OHCA outcomes are better when the resuscitation response is integrated. Incorporation of drone-delivered AEDs require an integrated response from multiple stakeholders, including resuscitation services, drone operators, regulators, call handlers, CPR training providers. I would cite this paper (DOI: 10.1016/j.resuscitation.2016.10.029), tying the positive outcomes to integration of resuscitation response.

Reviewer #2: This is a well-conducted and timely qualitative study exploring bystanders’ perspectives on drone-delivered AEDs in OHCA. It makes an important contribution to understanding public acceptance and barriers to implementation of emerging prehospital technologies.

Strengths:

Clear use of theoretical frameworks (TDF, COM-B, Behaviour Change Wheel) to guide analysis and intervention development.

Rich participant data, with well-chosen illustrative quotes.

Identification of practical intervention areas (call-handler scripts, training, co-responder models) with real-world applicability.

Transparent reporting following COREQ guidelines.

Suggestions for minor revision:

1) Data availability statement: Clarify that anonymised transcripts are available on request because of participant confidentiality, as PLOS requires explicit justification for restrictions.

2) Methods – saturation: Expand on how thematic saturation was determined (e.g., what criteria were used at 14 interviews to conclude no new insights were emerging).

3) Transferability of findings: Consider discussing how perceptions of drone technology may vary internationally and whether the UK-specific context (regulation, AED availability) limits generalisability.

4) Limitations: Expand slightly on the possible impact of self-selection bias (motivated or more confident bystanders may have been overrepresented) and the exclusion of suspected fraudulent respondents.

5) Language/clarity: Minor editorial corrections (e.g., sentence flow in Background and Discussion) will improve readability.

Overall, this is an important and well-executed manuscript that will interest both emergency medicine and health policy audiences. I recommend acceptance after minor revisions.

**Do you want your identity to be public for this peer review?** For information about this choice, including consent withdrawal, please see our Privacy Policy

Reviewer #1: No

Reviewer #2: **Yes: ** Jahed Hossain Nobel

---

## [Author Response · Author response to Decision Letter 1]

3 Nov 2025

Response to reviewers

Thank you for your review of the manuscript, and for the opportunity to resubmit. We hope the changes that we have made have resulted in an improved manuscript.

Comments to the Author

Have the authors made all data underlying the findings in their manuscript fully available?

Reviewer #1: No

Reviewer #2: No

Please note that we have now made anonymised transcripts available in the Supporting information submitted with this revision. We have clarified that the transcripts have been checked to make sure they were anonymised and we have further redacted any potentially identifiable information. We have clarified that participants were made aware that anonymised interview data could be shared in future, and that they provided explicit consent for this.

Reviewer #1:

The investigators explored public attitudes towards drone-delivered AEDs for OHCA by conducting 14 semi-structured interviews with real-life OHCA bystanders. They identified 5 themes and devised interventions.

1. The sample size is small and it is unclear whether the subjects are representative of the general population. Nevertheless, the study provides clues on the complexity of the process. I suggest that the authors acknowledge the small sample size in the limitations section.

We have added a section to the limitations

“The sample size in this study is small and the participants might not be representative of the general population. It may have been that more motivated or confident bystanders were more likely to put themselves forward as participants. Certainly, a higher proportion of our participants reported CPR/AED training than the general UK population [42].”

2. The availability of AED doesn’t guarantee it being used. I would cite the Home use of AED trial (HAT), which did not improve the OHCA outcomes (DOI: 10.1056/NEJMoa0801651). One of the reasons for failure was insufficient use of the AEDs, despite resuscitation training, including AED, provided to the caregivers in the study.

We agree that this is an important point. We have added the following text, including the above-mentioned reference:

“Further, there is no guarantee that an AED will be used if available, which has been demonstrated in studies reporting the use of AEDs in both public [39] and home [40] settings.”

3. OHCA outcomes are better when the resuscitation response is integrated. Incorporation of drone-delivered AEDs require an integrated response from multiple stakeholders, including resuscitation services, drone operators, regulators, call handlers, CPR training providers. I would cite this paper (DOI: 10.1016/j.resuscitation.2016.10.029), tying the positive outcomes to integration of resuscitation response.

Again, we agree that this is an important point. We have added the following text, including the suggested reference:

“It is important that any drone-delivery system that is implemented at scale is fully integrated into the community and ambulance service response to out-of-hospital cardiac arrest, which will ensure the best chance of improved patient outcomes [45].”

Reviewer #2:

This is a well-conducted and timely qualitative study exploring bystanders’ perspectives on drone-delivered AEDs in OHCA. It makes an important contribution to understanding public acceptance and barriers to implementation of emerging prehospital technologies.

Strengths:

Clear use of theoretical frameworks (TDF, COM-B, Behaviour Change Wheel) to guide analysis and intervention development.

Rich participant data, with well-chosen illustrative quotes.

Identification of practical intervention areas (call-handler scripts, training, co-responder models) with real-world applicability.

Transparent reporting following COREQ guidelines.

Thank you for your feedback.

Suggestions for minor revision:

1. Data availability statement: Clarify that anonymised transcripts are available on request because of participant confidentiality, as PLOS requires explicit justification for restrictions.

We have now made anonymised transcripts available in the supplementary material submitted with this revision.

2. Methods – saturation: Expand on how thematic saturation was determined (e.g., what criteria were used at 14 interviews to conclude no new insights were emerging).

Thank you, we have expanded the relevant section in the methods so that it now reads:

“We conducted 14 eligible interviews, at which point we determined we had obtained a sufficiently thorough and nuanced understanding of the topics presented [36]. This was determined by the interviewer (CJB) after discussion with CMS, and was based not only on no new codes being identified but also on agreement that we had a rich understanding of the topic at hand that was unlikely to be added to by additional recruitment.”

3. Transferability of findings: Consider discussing how perceptions of drone technology may vary internationally and whether the UK-specific context (regulation, AED availability) limits generalisability.

We have added a section in the limitations:

“Further, these findings may not be fully applicable outside the UK. Whilst bystanders in simulation studies internationally generally respond positively to drones delivering AEDs [17,21-23], the delivery model in the UK may vary from those in other countries. Reasons for this may include differences in aviation regulations, existing AED availability and ambulance response times. This potentially limits the applicability of our findings to other countries.”

4. Limitations: Expand slightly on the possible impact of self-selection bias (motivated or more confident bystanders may have been overrepresented) and the exclusion of suspected fraudulent respondents.

We have added the following limitations:

“The sample size in this study is small and the participants might not be representative of the general population. It may have been that more motivated or confident bystanders were more likely to put themselves forward as participants. Certainly, a higher proportion of our participants reported CPR/AED training than the general UK population [42].”

and (expanding our description of potential fraudulent responses):

“Nonetheless, determining whether or not participants were genuine was subjective [35], and so some uncertainty must remain about around the validity of other participants whose data has been analysed. If we have erred in our decisions to exclude participants based on suspected fraud, this may impact the validity of our results.”

5. Language/clarity: Minor editorial corrections (e.g., sentence flow in Background and Discussion) will improve readability.

We have reviewed the manuscript and made improvements where possible.

Overall, this is an important and well-executed manuscript that will interest both emergency medicine and health policy audiences. I recommend acceptance after minor revisions.

Thank you for your feedback.

---

## [Editor Report · Decision Letter 1]

5 Nov 2025

Bystanders’ attitudes towards drone delivered Automated External Defibrillators for out-of-hospital cardiac arrest: a qualitative interview study.

PONE-D-25-37331R1

Dear Dr. Smith,

We’re pleased to inform you that your manuscript has been judged scientifically suitable for publication and will be formally accepted for publication once it meets all outstanding technical requirements.

Kind regards,

Nik Hisamuddin Nik Ab. Rahman

Academic Editor

PLOS ONE
---

## [Editor Report · Acceptance letter]

PONE-D-25-37331R1

PLOS ONE

Dear Dr. Smith,

I'm pleased to inform you that your manuscript has been deemed suitable for publication in PLOS ONE. Congratulations! Your manuscript is now being handed over to our production team.

Kind regards,

on behalf of

Professor Dr Nik Hisamuddin Nik Ab. Rahman

Academic Editor

PLOS ONE